# Systematic proteomic analysis of LRRK2-mediated Rab GTPase phosphorylation establishes a connection to ciliogenesis

Martin Steger[1†], Federico Diez[2†], Herschel S Dhekne[3], Pawel Lis[2], Raja S Nirujogi[2], Ozge Karayel[1], Francesca Tonelli[2], Terina N Martinez[4], Esben Lorentzen[5], Suzanne R Pfeffer[3], Dario R Alessi[2*], Matthias Mann[1*]

[1]Department of Proteomics and Signal Transduction, Max-Planck-Institute of Biochemistry, Martinsried, Germany; [2]Medical Research Council Protein Phosphorylation and Ubiquitylation Unit, School of Life Sciences, University of Dundee, Dundee, United Kingdom; [3]Department of Biochemistry, Stanford University School of Medicine, Stanford, United States; [4]The Michael J. Fox Foundation for Parkinson's Research, New York, United States; [5]Department of Molecular Biology and Genetics, Aarhus University, Aarhus, Denmark

*For correspondence:
d.r.alessi@dundee.ac.uk (DRA);
mmann@biochem.mpg.de (MM)

†These authors contributed equally to this work

**Abstract** We previously reported that Parkinson's disease (PD) kinase LRRK2 phosphorylates a subset of Rab GTPases on a conserved residue in their switch-II domains (Steger et al., 2016) (PMID: 26824392). Here, we systematically analyzed the Rab protein family and found 14 of them (Rab3A/B/C/D, Rab5A/B/C, Rab8A/B, Rab10, Rab12, Rab29, Rab35 and Rab43) to be specifically phosphorylated by LRRK2, with evidence for endogenous phosphorylation for ten of them (Rab3A/B/C/D, Rab8A/B, Rab10, Rab12, Rab35 and Rab43). Affinity enrichment mass spectrometry revealed that the primary ciliogenesis regulator, RILPL1 specifically interacts with the LRRK2-phosphorylated forms of Rab8A and Rab10, whereas RILPL2 binds to phosphorylated Rab8A, Rab10, and Rab12. Induction of primary cilia formation by serum starvation led to a two-fold reduction in ciliogenesis in fibroblasts derived from pathogenic LRRK2-R1441G knock-in mice. These results implicate LRRK2 in primary ciliogenesis and suggest that Rab-mediated protein transport and/or signaling defects at cilia may contribute to LRRK2-dependent pathologies.
DOI: https://doi.org/10.7554/eLife.31012.001

## Introduction

LRRK2 encodes a large protein kinase and a number of LRRK2 mutations cause autosomal dominant Parkinson's disease (PD) (*Bardien et al., 2011*; *Funayama et al., 2005*; *Paisán-Ruíz et al., 2004*; *Zimprich et al., 2004*). Furthermore, genome-wide association studies (GWAS) have pinpointed LRRK2 as a risk factor for idiopathic PD, indicating that it is a master regulator of the molecular pathways controlling both hereditary and sporadic forms of PD (*Nalls et al., 2014*). LRRK2 is expressed at low levels in brain neurons and at higher levels in lung, kidney, pancreas and immune cells, therefore multiple cell types in different tissues might act in concert during PD pathogenesis (*Giesert et al., 2013*; *Thévenet et al., 2011*). The most frequent LRRK2 mutations that segregate with familial PD (R1441C, R1441G, Y1699C, G2019S and I2020T) all map to its catalytic domains, namely the ROC-COR (GTPase activity) and kinase domains (*Cookson, 2010*). The kinase domain mutations (G2019S and I2020T) increase kinase activity both in vitro and in vivo, and interestingly, R1441C, R1441G and Y1699C enhance this activity only in vivo (*Sheng et al., 2012*; *Steger et al., 2016*). This indicates that (i) the kinase and the GTPase domains communicate with each other and

(ii) LRRK2-dependent pathogenicity is triggered by an increase in kinase activity, regardless of the mutation position.

Multiple lines of evidence suggest that LRRK2 regulates intracellular vesicular traffic and that mutant LRRK2-associated trafficking defects contribute to PD pathogenesis (*Cookson, 2016*). First, LRRK2 translocates to membranes following Toll-like receptor stimulation of immune cells (*Schapansky et al., 2014*). Second, LRRK2 is reported to regulate the autophagosome/lysosome system, which is particularly interesting as defects in proteins coordinating this pathway are found in multiple forms of PD (*Beilina and Cookson, 2016*; *Giaime et al., 2017*; *Tang, 2017*). Third, LRRK2 phosphorylates at least three Rab GTPases (Rab8A, Rab10 and Rab12) in cultured human and murine cells as well as in mice, and this reduces their binding affinity for different Rab regulatory proteins (*Ito et al., 2016*; *Steger et al., 2016*; *Thirstrup et al., 2017*).

Rab GTPases are master regulators of intracellular vesicle trafficking and about 70 family members are expressed in humans (*Wandinger-Ness and Zerial, 2014*). In addition to the above-mentioned three Rabs that can be phosphorylated in a LRRK2-dependent manner, Rab29 (also known as Rab7L1) and Rab39B have been implicated in PD. Missense mutations in Rab39B lead to intellectual disability and PD-like clinical features (*Wilson et al., 2014*). Rab29 localizes to the trans-Golgi network (TGN) and Rab29 knockout mice mimic a phenotype of LRRK2 knockouts – an age-associated lysosomal defect in the kidney (*Baptista et al., 2013*; *International Parkinson's Disease Genomics Consortium et al., 2014*; *Kuwahara et al., 2016*). Our previous work showed that LRRK2 phosphorylates Rab8A, Rab10 and Rab12 at a conserved site in the switch II domain (Rab8A-T72, Rab10-T73 and Rab12-S106), which is conserved in 50 Rabs (*Steger et al., 2016*). We also found that LRRK2 specifically phosphorylates overexpressed Rab3A, but not Rab7A, despite their high-sequence homology adjacent to the site of phosphorylation. A prediction of whether or not a given Rab protein can be phosphorylated by LRRK2 in cells is thus not possible, based solely on the primary amino acid sequence. Although this work suggests that Rabs are involved in PD pathogenesis, the molecular mechanisms remain to be determined. Here, we systematically investigate whether Rabs other than 8A, 10 and 12 are LRRK2 targets. Subsequent modification-specific interaction experiments led us to establish an unsuspected effect of mutant LRRK2 on ciliogenesis via phosphorylated Rab GTPases.

## Results and discussion

### 14 overexpressed rab GTPases are phosphorylated by LRRK2

We set out to determine which of the 50 Rab proteins that contain a conserved, predicted LRRK2 target site are indeed phosphorylated by LRRK2. For this purpose, we overexpressed each of the 50 Rabs in triplicate in HEK293 cells, either alone or in combination with LRRK2 harboring the activating G2019S mutation (*Figure 1A* and *Figure 1—figure supplement 1*). To determine phosphorylation specificity, we included a third condition in which both kinase and substrate were expressed in the presence of the selective LRRK2 inhibitor, HG-10-102-01 (*Choi et al., 2012*). Although Rab32 and Rab38 do not contain the conserved phosphorylation site (*Figure 1—figure supplement 2A*), we included them because they have previously been reported to interact with LRRK2 (*Waschbüsch et al., 2014*).

Expression of all 52 constructs and inhibitor efficacy was demonstrated by immunoblotting (*Figure 1—figure supplement 2B*). To identify LRRK2 target candidates, we enriched the individual epitope-tagged Rab family members by immunoprecipitation before digestion, peptide purification and high-resolution, label-free LC-MS/MS analysis (*Figure 1A*). To generate peptides containing the phosphorylation site with an optimal length for MS analysis, we selected the most appropriate protease among trypsin, chymotrypsin, Asp-N and Glu-C (*Supplementary file 1*). This generated 468 LC-MS/MS files (52 Rabs under three conditions with triplicate analysis), which we stringently processed using the MaxQuant software for peptide identification and quantification (*Cox and Mann, 2008*). For each Rab, several phosphorylated peptides were generally detected, but only those containing the predicted site showed significant, quantitative differences as a result of LRRK2 overexpression and inhibition. In total, we quantified this site in 37 different Rab proteins. For 12 Rabs, the

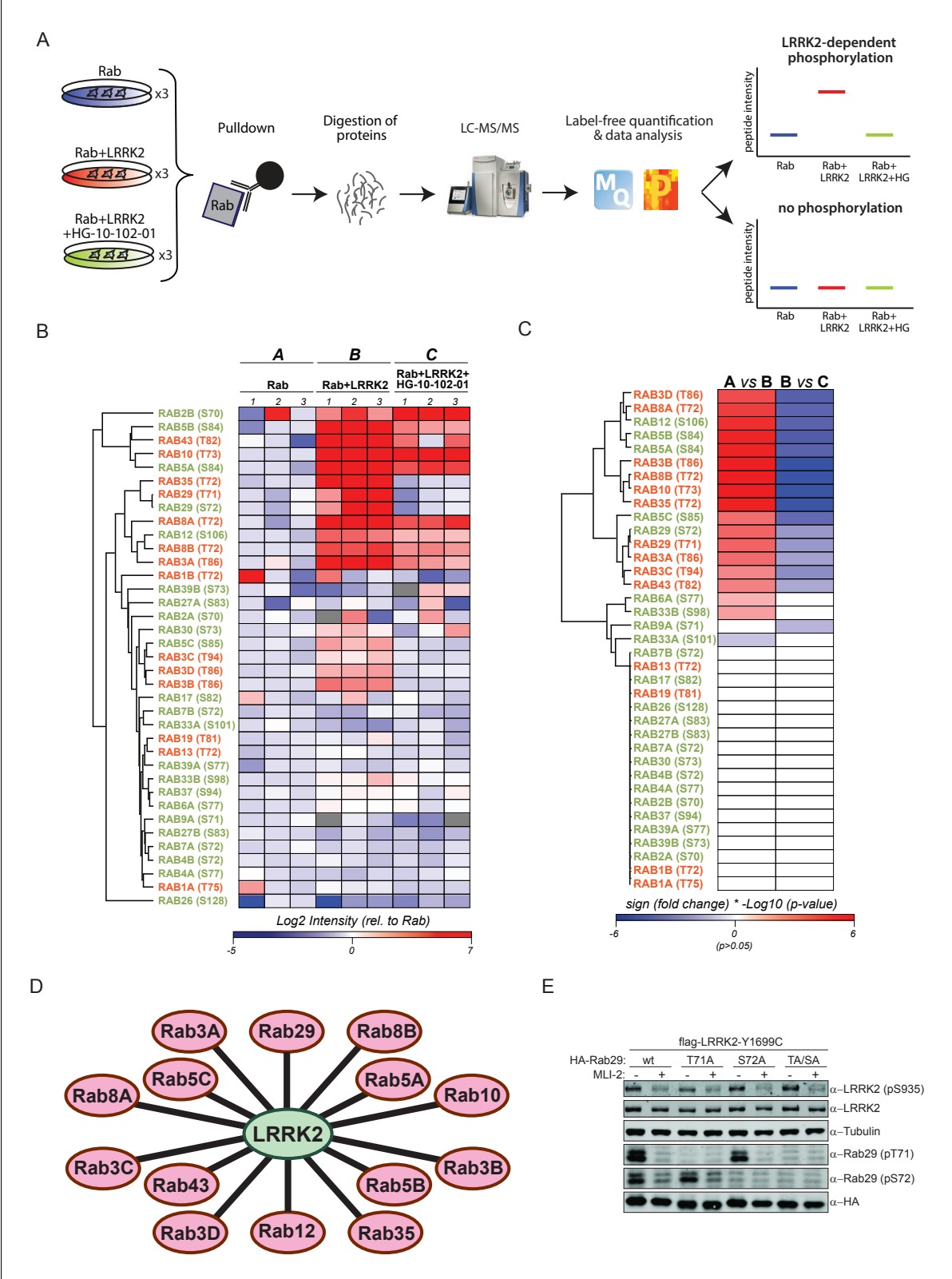

**Figure 1.** 14 Rab GTPases are phosphorylated by LRRK2 when overexpressed HEK293s. (**A**) Workflow for identifying Rabs that are phosphorylated by LRRK2 in an overexpression system. The individual Rab proteins were overexpressed alone or expressed in combination with active (G2019S)- or chemically inhibited LRRK2 (HG-10-102-01, 3 µM, 3 hr, n = 3). The Rabs were then affinity-enriched, digested and analyzed by label-free LC/MS-MS. Mass spectra database search and data analysis were performed using MaxQuant and Perseus, respectively. Substrates were scored when the

*Figure 1 continued*

predicted Rab phosphopeptide was detected, upregulated upon LRRK2 expression and reduced after inhibitor treatment. (B) Heat map of phosphopeptide intensities (Log2) and (C) Tukey adjusted p-values (-Log10, multiplied by the sign of the fold change of the respective group comparison [A = Rab, B = Rab + LRRK2, C = Rab + LRRK2 +HG-10-102-10]). Values missing in all replicates of one group were imputed and non-imputed, missing values are in grey. (D) Scheme of the 14 Rab proteins that are phosphorylated by LRRK2. (E) Western blot confirming that both Rab29-pT71 and Rab29-pS72 are phosphorylated by LRRK2-Y1699C when overexpressed in HEK293 cells.

DOI: https://doi.org/10.7554/eLife.31012.002

The following figure supplements are available for figure 1:

**Figure supplement 1.** Sequence alignment of 50 Rabs in which the predicted LRRK2 phosphorylation site is conserved.

DOI: https://doi.org/10.7554/eLife.31012.003

**Figure supplement 2.** Sequence alignment of Rab10, Rab38 and Rab32 and western blot analysis of 52 overexpressed Rab GTPases.

DOI: https://doi.org/10.7554/eLife.31012.004

**Figure supplement 3.** Both Rab29-T71 and Rab29-S72 are phosphorylated by overexpressed LRRK2 in HEK293 cells.

DOI: https://doi.org/10.7554/eLife.31012.005

predicted phosphopeptide was not found. Nonetheless, we were able to readily detect and quantify the non-phosphorylated counterpart, suggesting that the modified form should have been detected if it had been present. Thus, we conclude that in these cases, the site is either not phosphorylated at all, or the phosphorylation stoichiometry is below the detection limit. The LC-MS/MS analysis of the remaining Rabs - Rab6C, Rab9B and Rab41 – was inconclusive as we were not able to either identify the phosphorylated- or the non-phosphorylated predicted LRRK2 target peptides (*Supplementary file 1*).

The proteolytic cleavage of some Rabs with very high-sequence homology (e.g. Rab3A/B/C/D) creates peptides with identical primary amino acid sequence, making them indistinguishable to MS and MS/MS. However, as we individually overexpressed and immunoprecipitated Rabs with mono-clonal antibodies directed against the epitope tag, it is very unlikely that several such proteins are enriched at equal efficiencies. Our analysis revealed 14 phosphopeptides that were regulated in a LRRK2-dependent manner, in that their abundance was significantly increased upon kinase expression and decreased when cells were treated with HG-10-102-01 (Tukey's multiple comparisons test, $p < 0.05$) (*Figure 1B–D*). This includes all the previously identified LRRK2 substrates: Rab8A, Rab10 and Rab12 (*Steger et al., 2016*), and adds Rab3A/B/C/D, Rab5A/B/C, Rab8B, Rab29, Rab35 and Rab43.

Rab29 was recently shown to interact with LRRK2, both physically and genetically, and Rab29 knockout in mice phenocopies LRRK2 knockouts, indicating that both proteins might act in the same pathway (*International Parkinson's Disease Genomics Consortium et al., 2014*; *Kuwahara et al., 2016*; *Simón-Sánchez et al., 2009*). Interestingly, we identified a doubly-phosphorylated peptide (pT71 and pS72) that was regulated by LRRK2. To confirm this finding, we individually mutated both residues to non-phosphorylatable alanine and expressed the constructs in HEK293 cells, along with active or chemically inhibited LRRK2. We quantified the levels of the peptides with either alanine substitution by MS and found that the regulation persisted in each case (*Figure 1—figure supplement 3*). Finally, we raised two phospho-specific antibodies recognizing either Rab29-pT71 or Rab29-pS72, which further substantiated the evidence that both residues are modified by LRRK2 in this system (*Figure 1E*). Interestingly, the 14 Rabs that are phosphorylated by LRRK2 are widely dispersed over the Rab phylogenetic tree and further analysis is required to understand the determinants of phosphorylation. It is likely that they co-localize with LRRK2 and this may account for the phosphorylation specificity.

## LRRK2 phosphorylates at least ten endogenous Rab GTPases in cells

Because pathogenic LRRK2 mutations occurring in different functional domains increase the phosphorylation of Rab8A, Rab10 and Rab12 in cells (*Ito et al., 2016*; *Steger et al., 2016*), we next investigated if the 14 Rabs regulated in our screen were equally affected by mutations in LRRK2. For this, we used the phos-tag assay, a method complementary to MS, in which phosphorylated Rabs are resolved from their non-phosphorylated counterparts by SDS-PAGE (*Ito et al., 2016*). As judged by the increased Rab-specific band-shift intensity, all LRRK2 pathogenic mutations analyzed (G2019S, R1441G, Y1699C) led to increased phosphorylation of 13 Rab substrates (*Figure 2A*). For

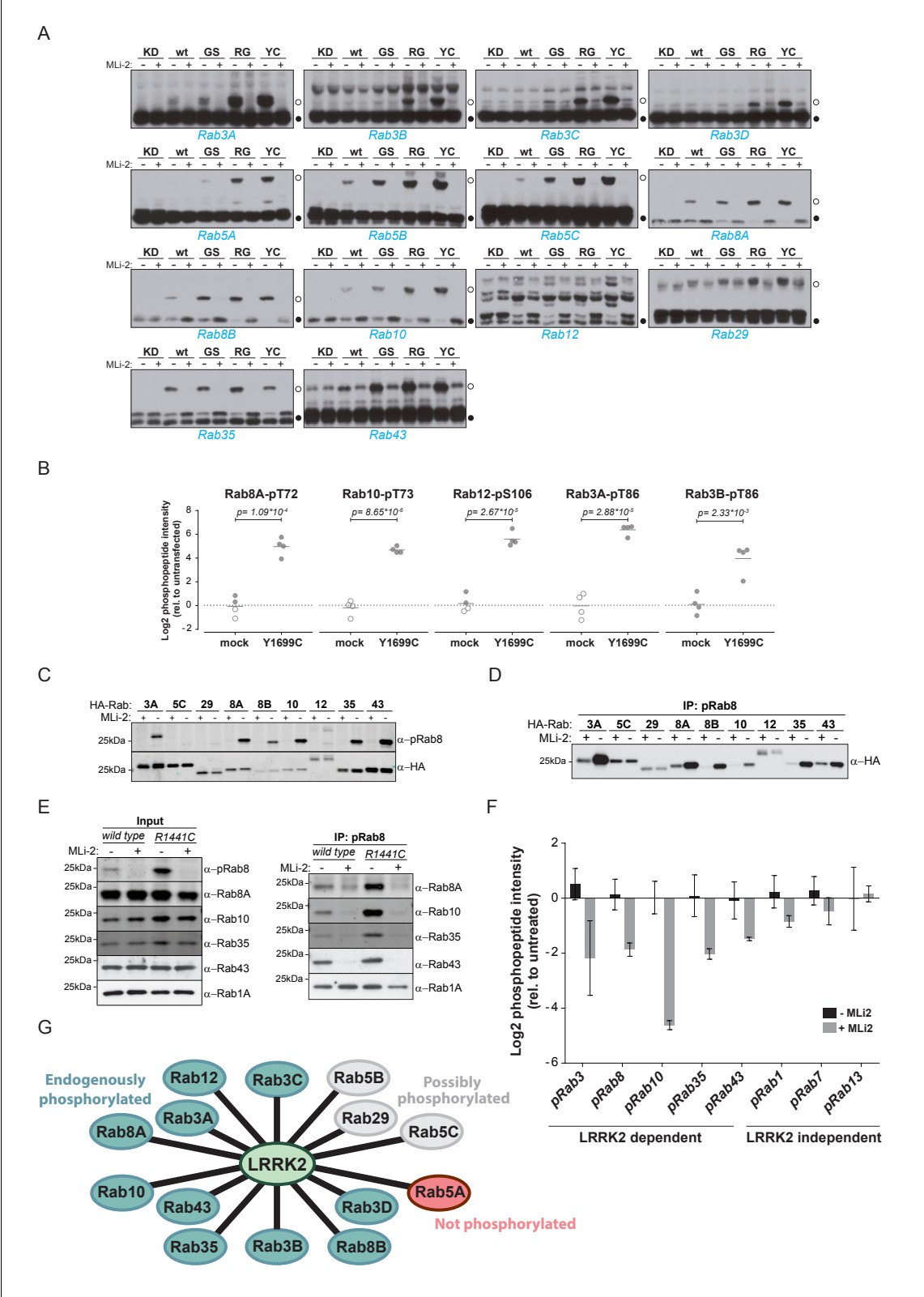

**Figure 2.** Ten endogenous Rabs are phosphorylated by LRRK2 in cells. (**A**) Phos-tag SDS-PAGE of 14 Rabs co-expressed with different LRRK2 variants (KD = kinase dead, wt = wild type, GS= G2019S, RG = R1441G, YC= Y1699C) in HEK293 cells. Filled circles indicate unmodified Rabs and open circles, phosphorylated forms (MLi-2 = 100 nM, 1 hr). (**B**) MS-quantified Rab3A-pT86, Rab3B-pT86, Rab8A-pT72, Rab10-pT73 and Rab12-pS106 peptide intensities from mock and LRRK2-Y1699C transfected cells. Open circles indicate imputed values. (**C**) Western blot analysis of HA-tagged Rabs co-

*Figure 2 continued on next page*

*Figure 2 continued*

expressed with LRRK2-Y1699C in HEK293 cells (-/+100 nM MLi-2, 2 hr) using polyclonal anti-phospho-Rab8 and HA antibodies. (D) Same as (C) but Rabs were immunoprecipitated using the anti-pRab8 antibody and HA antibody was used for detection. (E) Western blot of endogenous Rabs in wild type and LRRK2-R1441C MEFs (-/+100 nM MLi-2, 1 hr) using the indicated antibodies (left) and immunoprecipitation of Rabs using anti-pRab8 antibody (right). (F) MS-quantified Rab phosphosite intensities of pRab8 immunoprecipitation in R1441C MEFs (-/+100 nM MLi-2, 2 hr). Error bars are mean ±SEM (n = 3). (G) 10 Rab proteins at endogenous expression levels are phosphorylated by LRRK2 in cells (green). Rab5A is not (red) and Rab5B, Rab5C and Rab29 are possibly phosphorylated (grey).

DOI: https://doi.org/10.7554/eLife.31012.006

The following figure supplements are available for figure 2:

**Figure supplement 1.** Rab5A was immunoprecipitated from mock or LRRK2-Y1699C-transfected HEK293 cells.

DOI: https://doi.org/10.7554/eLife.31012.007

**Figure supplement 2.** Nine Rab GTPases containing a Thr site in their switch-II domain are phosphorylated by LRRK2.

DOI: https://doi.org/10.7554/eLife.31012.008

Rab12, the assay resulted in multiple bands and no meaningful interpretation of the migration pattern was possible; however, our previous work had already established this protein as a LRRK2 substrate. In this context, we also investigated Rab6C, Rab9B and Rab41 by the phos-tag method, for which the MS analysis was inconclusive, and found no evidence of LRRK2-induced mobility shifts.

We next investigated which of these 14 Rab proteins are phosphorylated by LRRK2 endogenously. For this, we overexpressed the pathogenic LRRK2-Y1699C form in HEK293 cells, immunoprecipitated endogenous Rab proteins using commercially available- (Rab3A, Rab3B, Rab5A, Rab8A, Rab10) or in-house raised antibodies (Rab12) and quantified LRRK2-mediated phosphorylation by LC-MS/MS. As expected from our previous results, Rab8A-pT72, Rab10-pT73 and Rab12-pS106 phosphopeptide levels were strongly upregulated (~16 fold) in cells expressing LRRK2-Y1699C (*Figure 2B*). Rab3A-pT86 was also clearly regulated by LRRK2 in this system, but we were not able to detect any phosphopeptides mapping to either Rab3B-pT86 or Rab5A-pS84, despite the fact that the corresponding unmodified peptides were sequenced multiple times and our initial overexpression screen had established detectability of these phosphopeptides. Therefore, to further increase mass spectrometric sensitivity, we used stable isotope labeled (SIL) peptides equivalent to the endogenous Rab peptides of interest to guide targeted quantification. In contrast to the absolute quantification (AQUA) strategy (*Gerber et al., 2003*) that relies on fragment ions for quantification, we recorded both light and heavy counterparts in a multiplexed selected ion monitoring (SIM) scan mode on a quadrupole Orbitrap analyzer and used the heavy form to indicate the retention time (heavy) and the light form for quantification. This combination of the spike-in SIL peptide together with a very sensitive MS method should allow quantification of very low abundant endogenous Rab phosphorylation (Materials and methods). With this increased sensitivity, we found LRRK2-specific modification of Rab3B, but still failed to observe clear LRRK2-mediated phosphorylation of Rab5A-S84 (*Figure 2B* and *Figure 2—figure supplement 1*). Thus, Rab5A phosphorylation by LRRK2 in HEK293 cells occurs only when both the Rab and kinase are overexpressed exogenously. It is possible that accessory factors in other cell types are required for efficient LRRK2 phosphorylation of this protein.

For the remaining Rab proteins for which no suitable antibodies for total-Rab protein enrichment were available, we raised a rabbit polyclonal Rab antibody against the LRRK2-phosphorylated Rab8 switch-II motif. This efficiently recognized and immunoprecipitated LRRK2 phosphorylated Rabs containing a Thr site: namely, Rab3, Rab8, Rab10, Rab35 and Rab43 (*Figure 2C,D*). As an exception, the antibody failed to recognize pRab29 or to immunoprecipitate LRRK2-phosphorylated Rab29 protein. This could be either due to the particular linear sequence surrounding T71 (−1 position compared to the equivalent site on other Rabs) or due to double phosphorylation (T71 and S72) by LRRK2. Next, we treated wild type and LRRK2-R1441C knock-in mouse embryonic fibroblasts (MEFs) with the selective LRRK2 inhibitor MLi-2 (*Fell et al., 2015*), subjected the cell lysates to immunoprecipitation with the phospho-Rab8 antibody and analyzed the immunoprecipitates by immunoblotting or LC-MS/MS. Immunoblotting of the input cell extracts confirmed that compared to wild type, the signal detected by the newly developed anti-phospho-Rab8-antibody strongly increased in LRRK2-R1441C MEFs, and this was blocked by MLi-2 treatment (*Figure 2E*). For detecting immunoprecipitates by western blotting, we used CRISPR/CAS9 knock-out-validated antibodies (*Figure 2—figure*

*supplement 2A*). This revealed specific enrichment of phospho-Rab8A, -Rab10, -Rab35 and -Rab43, which increased in LRRK2-R1441C MEFs and was completely prevented by MLi-2 treatment (*Figure 2E*). In contrast, Rab1A phosphorylation was not sensitive to MLi-2 in this system, confirming the data from our initial overexpression screen.

Using the anti-phospho-Rab8 antibody in LRRK2-R1441C fibroblasts, we analyzed immunoprecipitates by LC-MS/MS. Quantification of 32 Rab protein intensities showed that Rab3A, Rab8A, Rab8B, Rab10, Rab35 and Rab43 were enriched more than twofold in a LRRK2-dependent manner. Rab3B and Rab3D were also detected, however, compared to MLi-2 treatment, their enrichment in untreated cells was less pronounced (about 1.5-fold), probably due to lower protein expression levels (*Figure 2—figure supplement 2B,C*). In concordance with this, quantification of the switch-II phosphorylated peptides of Rab3, Rab8, Rab10, Rab35 and Rab43 in the same experiment revealed a more than two-fold decrease upon MLi-2 treatment (*Figure 2F*). For Rab1A, Rab7 and Rab13, however, there was no pronounced difference in the corresponding protein and phospho-peptide levels when comparing untreated with MLi-2-treated samples, suggesting that these Rabs are phosphorylated by a kinase other than LRRK2. Finally, we immunoprecipitated phosphorylated Rabs from HEK293 cells expressing LRRK2-Y1699C using the same antibody. Again, we confirmed LRRK2-specific enrichment of nine threonine phosphorylated Rab proteins. These were Rab3A/B/C/D, Rab8A/B, Rab10, Rab35 and Rab43 and the protein sequence coverage ranged from 33% (Rab3B) to 90% (Rab10). Compared to the other family members, this indicates that they are recognized by the antibody in a direct manner (*Figure 2—figure supplement 2D*).

In summary, our study demonstrates that 14 Rab proteins are phosphorylated by LRRK2 at the switch-II site in an overexpression system (Rab3A/B/C/D, Rab5A/B/C, Rab8A/B, Rab10, Rab12, Rab29, Rab35 and Rab43). For ten of these (Rab3A/B/C/D, Rab8A/B, Rab10, Rab12, Rab35 and Rab43), endogenous LRRK2-dependent phosphorylation is clearly established; for one (Rab5A), we observed phosphorylation in an overexpression but not an endogenous setting. In vivo analysis of the remaining three Rabs (Rab5B, Rab5C and Rab29) was hampered by the unavailability of suitable antibodies for protein enrichment or detection (*Figure 2G*).

## Phosphorylation-dependent binding of regulatory proteins to Rabs

Different residues in the switch-II domain of Rabs can be modified post-translationally, and this interferes with binding to partner proteins such as Rab GDP dissociation inhibitor alpha/beta (GDI1/2). For example, Rab1B is AMPylated on Y77 (Y80 in Rab1A) by the Legionnaires' disease protein, DrrA, resulting in a constitutively active Rab protein that fails to interact with GTPase-activating proteins (GAPs) and GDIs (*Müller et al., 2010*; *Oesterlin et al., 2012*). Similarly, the switch-II domain of Rab1A can be phosphocholinated on S79 during *Legionella* infection, which results in strongly decreased GDI binding (*Goody et al., 2012*; *Mukherjee et al., 2011*). We showed previously that phosphorylation of Rab8A-T72 and the equivalent sites, Rab10-T73 and Rab12-S106 by LRRK2 decreases GDI binding (*Steger et al., 2016*). Substitution of Rab7A-T72 with a phosphomimetic glutamic acid has been shown to abrogate GDI interaction (*Satpathy et al., 2015*). Here, we set out to test systematically, the effect of replacing the predicted LRRK2 target site with a negatively charged, phosphomimetic glutamic acid residue on partner protein binding. For this, we mutated these sites to either non-phosphorylatable alanine or glutamic acid in all 14 Rabs (both T71 and S72 for Rab29) and expressed them in HEK293 cells (n = 3). Following affinity-enrichment, we digested bound proteins and analyzed the resulting peptides by label-free LC-MS/MS. As expected, the S/T→E mutations strongly reduced partner binding for all tested Rabs. The S/T→A Rab mutants instead stably bound to GDI1/2 and the Rab escort proteins, CHM and CHML, except for both tested Rab29 constructs (*Figure 3—figure supplement 1A,B*). In that case, neither the T71A/E nor the S72A/E constructs interacted with GDI, indicating either low binding affinities, an alteration in nucleotide-binding properties, or protein misfolding.

To investigate more specifically the effect of LRRK2-induced Rab phosphorylation on protein interactions, we expressed LRRK2 in HEK293 cells and determined the interactomes of both HA-tagged and endogenous Rab8A by AE-MS, before and after chemical LRRK2 inhibition. To increase phosphosite occupancy at T72, we used the pathogenic ROC-COR domain LRRK2-R1441G mutant, which confers strong intracellular kinase activity (*Sheng et al., 2012*; *Steger et al., 2016*). Upon LRRK2 inhibition, Rab8A-pT72 levels decreased about eight fold as shown by quantitative MS, establishing inhibitor efficacy (*Figure 3—figure supplement 2*). As we had reported previously, GDIs

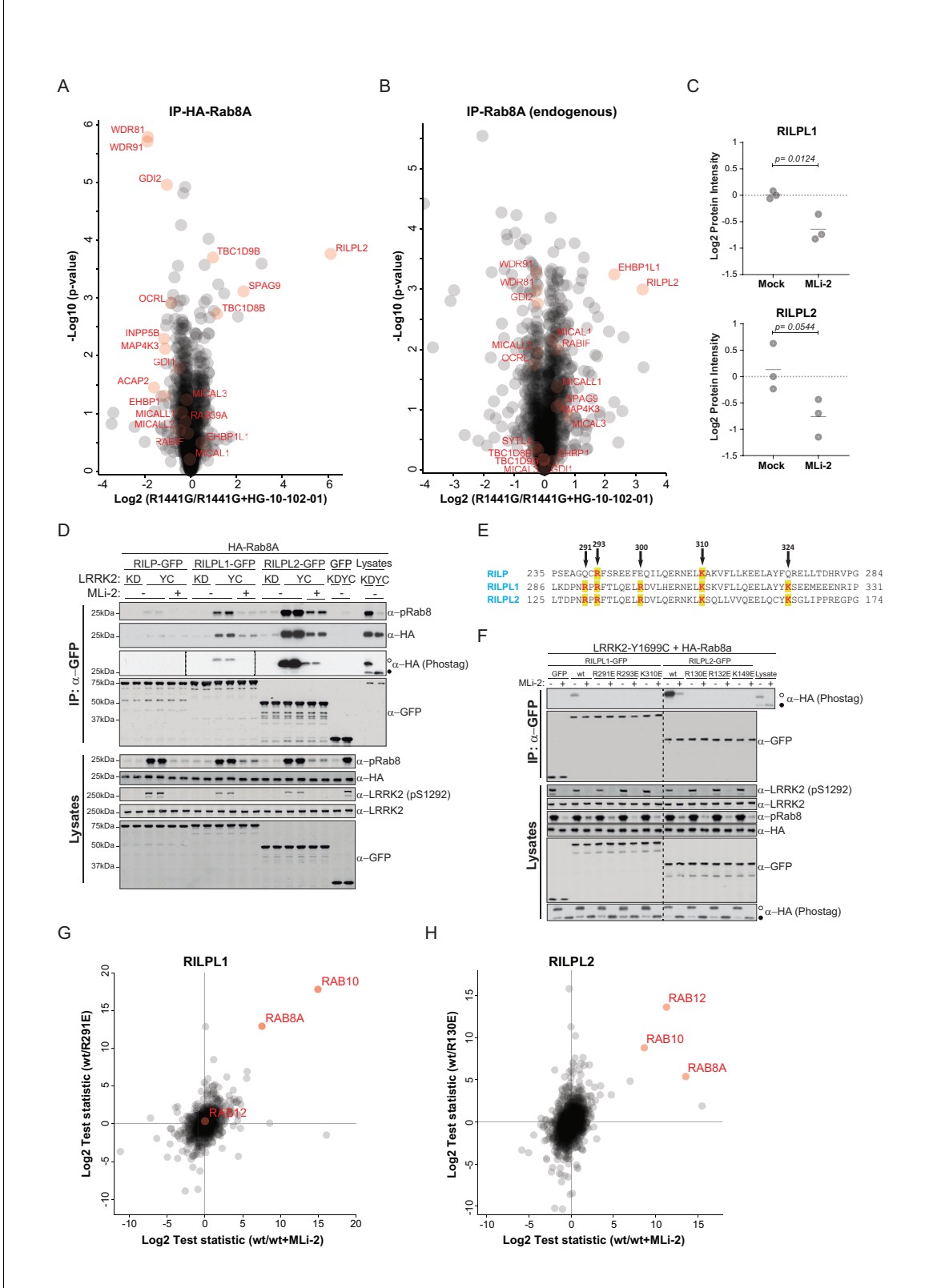

**Figure 3.** Phosphorylation-specific protein binding to Rabs. (**A**) AE-MS of HA-Rab8A and (**B**) endogenous Rab8A using extracts of GFP-LRRK2-R1441G expressing Flp-In T-Rex HEK293 cells. Expression of the kinase was induced for 48 hr by doxycycline (1 μg/ml) and LRRK2 inhibited using HG-10-102-01 (3 μM, 3 hr). LRRK2-regulated, known and unknown Rab8A-binding proteins in both AE-MS experiments are highlighted in red. (**C**) Label-free (LFQ [*Cox et al., 2014*]), MS-quantified RILPL1 and RILPL2 levels after immunoprecipitation of Rab8A from mock- or MLi-2 (200 nM, 2 hr) treated LRRK2-

*Figure 3 continued on next page*

*Figure 3 continued*

R1441C MEFs. (D) Pulldown of GFP-tagged RILP, RILPL1 or RILPL2, transiently expressed with HA-Rab8A and the indicated LRRK2 variants (KD= Y1699C/D2017A) in HEK293 cells. Western blot after Phos-tag SDS-PAGE was used to detect interacting proteins using the indicated antibodies. (MLi-2 = 150 nM, 2 hr). (E) Sequence alignment of the RILP homology (RH) domains of RILP, RILPL1 and RILPL2 showing five conserved basic residues, which are highlighted. (F) Same as (D) but using different RILPL1 and RILPL2 mutants. For phos-tag blots, filled circles indicate non phosphorylated proteins and open circles phosphorylated proteins. (G) AE-MS of GFP-RILPL1 (wt or R291E) and (H) GFP-RILPL2 (wt or R130E), expressed with LRRK2-Y1699C in HEK293 cells, and treated or not with MLi-2 (for wt, 200 nM, 2 hr). The student's two sample test statistic (Log2) of the indicated comparisons was used for plotting.

DOI: https://doi.org/10.7554/eLife.31012.009

The following figure supplements are available for figure 3:

**Figure supplement 1.** Phosphomimetic S/T->E mutation of the LRRK2 phosphorylation site abrogates GDI1/2 and CHM/CHML binding in 13 Rab GTPases.

DOI: https://doi.org/10.7554/eLife.31012.010

**Figure supplement 2.** MS-quantified pT72-Rab8A peptide levels in HEK293 cells expressing LRRK2-R1441G.

DOI: https://doi.org/10.7554/eLife.31012.011

**Figure supplement 3.** Phospho-dependent interaction of Rab8A, Rab10 and Rab12 with RILPL1/2.

DOI: https://doi.org/10.7554/eLife.31012.012

associated preferentially with the non-phospho forms of both HA-Rab8A and endogenous Rab8A (*Figure 3A,B*). However, this effect was more pronounced in the case of overexpressed Rab8A, probably due to the higher fraction of phosphorylated protein.

Several, to-date undescribed interaction partners that showed preferential binding to the non-phospho-forms in both the endogenous and HA-tagged systems (*Figure 3A,B*). These include the poorly characterized, WD repeat domain (WDR)-containing proteins, WDR81 and WDR91, which have recently been linked to regulation of endosomal phosphatidylinositol 3-phosphate levels in *C. elegans* and mammalian cells (*Liu et al., 2016*). The inositol-5-phosphatase, OCRL is a known Rab8A effector, and our experiments revealed that phosphorylation abolishes its binding to Rab8A. Mutations in the human OCRL gene cause the oculocerebrorenal syndrome of Lowe and it would be interesting to investigate whether the phosphorylation of Rabs is relevant in this context (*Hou et al., 2011*).

Strikingly, we found that Rab interacting lysosomal protein like 2 (RILPL2) binds preferentially to the phosphorylated forms of both HA-Rab8A and Rab8A. RILPL2 contains a C-terminal, coiled-coil Rab-binding domain, which is known as RILP homology (RH) domain, and this is conserved in four other members of this protein family (RILP, RILPL1 and JIP3/4) (*Matsui et al., 2012*; *Wu et al., 2005*). RILPL1 and RILPL2 are poorly characterized proteins; however, a recent report links both to the regulation of protein localization in primary cilia (*Schaub and Stearns, 2013*). RILP is much better studied, and interacts with Rab7 to control lysosomal motility in cells (*Jordens et al., 2001*).

To investigate Rab8A-RILPL2 interaction in an endogenous setting, we immunoprecipitated Rab8A from mock- or MLi-2 treated, knock-in MEFs expressing the hyperactive LRRK2 variant, R1441C. This confirmed our initial LRRK2-specific interaction of Rab8A and RILPL2 and also identified RILPL1 as phospho-dependent Rab8A interactor (*Figure 3C*).

To further verify the pRab8A interaction with RILPL1 and RILPL2 and to test for RILP interaction, we co-expressed GFP-tagged, full length RILP, RILPL1 and RILPL2 with HA-Rab8A and LRRK2-Y1699C or kinase-dead LRRK2-Y1699C/D2017A. Both RILPL1 and RILPL2, but not RILP, specifically bound to LRRK2-generated phospho-Rab8A, and this interaction was strongly reduced by MLi-2 (*Figure 3D*). In this side by side comparison, RILPL2-bound phosphorylated Rab8A much stronger than RILPL1, indicating that RILPL2 might be a key effector specific for pRab8A. Phos-tag analysis independently confirmed that only the LRRK2-phosphorylated HA-Rab8A protein was co-immunoprecipitated with either RILPL2 or RILPL1 (*Figure 3D*).

There are five highly conserved basic residues in the RH domain on RILPL1 and RILPL2 that could potentially mediate the interaction with LRRK2-phosphorylated Rab8A (R291, R293, R300, K310 and K324 in RILPL1) (*Figure 3E*). Individual mutation of R291, R293 or K310, but not R300 or K324, abolished the interaction of N-terminally truncated RILPL1 with LRRK2-phosphorylated Rab8A (*Figure 3— figure supplement 3A*). Accordingly, the identified mutations abrogated pRab8 interaction with both full-length RILPL1 and RILPL2 (*Figure 3F*).

To extend our RILPL1/2-Rab interaction analysis reciprocally, we expressed GFP-RILPL1 (wt or R291E) and GFP-RILPL2 (wt or R130E) with LRRK2-Y1699C in HEK293 cells, in the presence or absence of MLi-2, and processed the samples for LC-MS/MS analysis after affinity enrichment. Strikingly, in this setup, RILPL1-wt interacted not only with Rab8A, but also with Rab10, and this was strongly reduced by MLi-2 and the R291E Rab interaction mutant (*Figure 3G*). RILPL2 instead bound all three of our previously identified bona-fide LRRK2 substrates, Rab8A, Rab10 and Rab12 in a LRRK2-dependent manner (*Steger et al., 2016*) (*Figure 3H*). We further confirmed the phospho-specific Rab8A, Rab10 and Rab12 interactions with RILPL1/2 by immunoprecipitation-phos-tag and phospho-Rab directed antibodies (*Figure 3—figure supplement 3B–D*). Together these results unambiguously establish that both RILPL1/2 are phospho-specific Rab interactors and that this interaction is mediated by the RH domain of RILPL1 and RILPL2. Based on these findings, it will be interesting to investigate possible genetic association of RILPL1/2 mutations with PD.

## Ciliogenesis is attenuated in cells expressing hyperactive LRRK2

Primary cilia are hair-like structures extending from the cell body and are found on nearly all human cells (*Malicki and Johnson, 2017*). Ciliary defects lead to a variety of human diseases known as ciliopathies, some more severe than others, and more than 200 genes have been associated with these disorders (*Reiter and Leroux, 2017*). Rab or Rab-regulatory protein dysfunctions play important roles in a number of ciliopathies. For instance, Rab28 is localized at the periciliary membrane of ciliated neurons and mutations in its gene are associated with cone-rod dystrophy, an inherited ocular disorder (*Jensen et al., 2016*; *Roosing et al., 2013*). Similarly, Rab23 is involved in ciliary protein transport and Rab23 mutations are associated with a rare congenital disorder known as Carpenter syndrome (*Boehlke et al., 2010*; *Jenkins et al., 2007*). Rab8A, Rab10, Rab11, Rab17 and Rab29 are involved in ciliogenesis; however, no specific disorders are associated with mutations in these genes (*Knödler et al., 2010*; *Nachury et al., 2007*; *Onnis et al., 2015*; *Sato et al., 2014*; *Yoshimura et al., 2007*). Since LRRK2 phosphorylation of Rab8A, Rab10 and Rab12 results in increased RILPL1/2 interaction (*Figure 3*) and Rab8A, Rab10 and RILPL1/2 are reported to regulate primary ciliogenesis (*Nachury et al., 2007*; *Sato et al., 2014*; *Schaub and Stearns, 2013*), we reasoned that LRRK2 itself might be a regulator of primary cilia formation. To test this, we induced cilia formation by serum starvation in LRRK2-R1441G knock-in MEFs that harbor increased kinase activity, and cultured them with or without MLi-2. Strikingly, as judged by anti-Arl13B staining, DMSO-treated LRRK2-R1441G expressing cells displayed a twofold decrease in the number of ciliated cells and a reduction in cilia length compared with overnight-MLi-2-treated cells (*Figure 4A–C*). To confirm this finding using another cell system, we analyzed cilia formation in 3T3 fibroblasts transfected with GFP-LRRK2-G2019S. In this case, about 30% of serum-starved, LRRK2-expressing cells (GFP positive) were ciliated (Arl13B positive). Overnight LRRK2 inhibitor treatment (MLi-2) increased this by twofold, demonstrating kinase dependency of the ciliation defect (*Figure 4D,E*). Thus, two different cellular assays support our finding that LRRK2 regulation is important for ciliogenesis. To determine whether the observed defect in ciliogenesis seen in hyperactive LRRK2 mutant-expressing cells is due to Rab8A phosphorylation, we expressed GFP-Rab8A-T72A or GFP-Rab8A-T72E in serum-starved RPE-1 cells and assessed both primary cilia formation and exogenous Rab8A protein localization by immunofluorescence microscopy. While GFP-Rab8A-wt was localized at cilia in about 80% of Arl13B-positive cells, this was not true for the T72A nor the T72E mutants, and total numbers of ciliated cells did not change in any of the conditions analyzed (*Figure 4F–H*). Thus, exogenous expression of the T72A or T72E mutants alone was not sufficient to block cilia formation. This suggests that other LRRK2 targets might play a role in this process or that endogenous Rab8A is sufficient to sustain cilia formation in this experimental setup.

Together, our data demonstrate that LRRK2 activity influences ciliogenesis and kinase hyperactivity interferes with this process. The precise details of how LRRK2-mediated Rab8A phosphorylation controls this process and influences RILPL1 and RILPL2 function will be important areas for future work. Whether phosphorylation of Rabs other than Rab8A by LRRK2 influences cilia formation will also require further analysis.

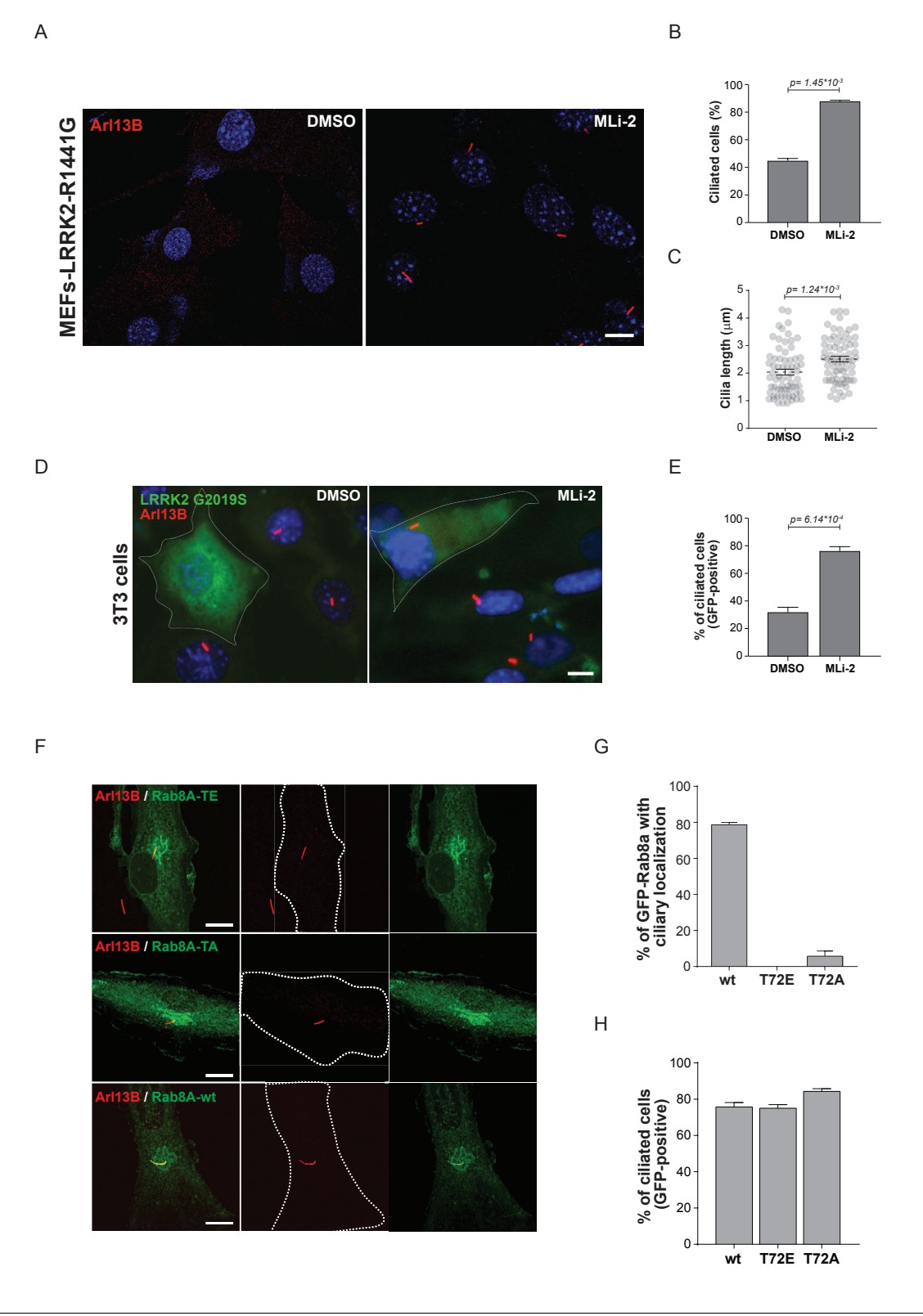

**Figure 4.** Pathogenic LRRK2 mutations inhibit primary cilia formation. (**A**) LRRK2-R1441G knock-in MEFs were serum starved overnight and treated with 200 nM MLi-2 (right) or DMSO (left). Primary cilia were stained using mouse anti-Arl13B (red) and nuclei using DAPI. (**B**) Quantification of primary cilia (Arl13B staining, n = 2,>100 cells per condition) and (**C**) cilia length (70 per condition). Scale bar = 10 µm. (**D**) NIH3T3 cells transfected with eGFP-LRRK2-G2019S were serum starved for 24 hr in the presence or absence of MLi-2 (200 nM). Scale bar = 10 µm. (**E**) Quantification of primary cilia (Arl13B

*Figure 4 continued on next page*

*Figure 4 continued*

positive) in GFP-positive cells;>50 cells per replicate were counted (n = 3). Error bars represent SEM and p-values were determined using unpaired, two-tailed Student's t-tests. (F) RPE-1 cells were infected with a lentivirus encoding GFP-Rab8A (T72E, T72A or wt) and 48 hr later, serum starved overnight to induce ciliation; primary cilia were detected with mouse anti-Arl13B (red). Dotted lines indicate cell outlines. Scale bars = 10 μm. (G) Quantification of cells with ciliary (Arl13B positive) GFP-Rab8A localization; (H) Quantitation of primary cilia (Arl13B staining) in GFP-positive cells. Error bars represent SEM from three experiments with >75 cells counted per condition.

DOI: https://doi.org/10.7554/eLife.31012.013

## Conclusion

Our systematic analysis of LRRK2-dependent Rab phosphorylation revealed that a distinct subset of the Rab family is subject to regulation by this kinase. When overexpressed a total of 14 Rabs are phosphorylated by LRRK2, and at least ten are phosphorylated when they are present in cells at endogenous levels. New tools such as specific antibodies or sensitive targeted MS methods, as well as more in depth analyses of tissues expressing LRRK2, are needed to clarify whether the remaining three Rabs (Rab5B/C and Rab29) are true endogenous LRRK2 substrates. Phosphomimetic substitution of the identified LRRK2 phosphosites abrogated the binding of Rabs to GDI1/2 and CHM/CHML, demonstrating that the phosphorylation of the switch-II domain is a general regulatory mechanism of the Rab cycle. Our data suggest that kinases other than LRRK2 phosphorylate the switch-II domain of some Rabs in different contexts. In fact, we found 37 overexpressed Rabs to be phosphorylated within this domain, but only 14 of these are catalyzed by LRRK2. Recent reports that Rab7A is phosphorylated on T72 during B cell signaling and that Rab1A-T75 is a TAK1 target (*Levin et al., 2016*; *Satpathy et al., 2015*) support this proposal.

Interactome analysis of all LRRK2-regulated Rabs confirmed GDI-binding as a general feature of the non-phospho forms. Unexpectedly, our experiments revealed different binding partners when the phosphomimetic S/T→E substitution was compared with LRRK2-induced phosphorylation. This suggests that the S/T→E mutation does not accurately substitute for phosphorylation in this context. It will therefore be necessary to analyze systematically, phosphorylated, endogenous Rabs to identify proteins that bind specifically to their phosphorylated forms. The fact that Rabs are only phosphorylated to a low extent in cells may pose challenges for the detection of modification-specific interactors.

Our previous work suggested that LRRK2 phosphorylation of Rab proteins leads to their functional inactivation due to interference with binding to specific effectors and their regulating proteins (*Steger et al., 2016*). Here, we have discovered a dominant interaction whereby phosphorylated Rabs bind preferentially to RILPL1 and RILPL2, proteins that are key for ciliogenesis (*Schaub and Stearns, 2013*). We verified this phosphorylation-dependent interaction in a variety of assays and showed that RILPL1 and RILPL2 are important binding partners of LRRK2 phosphorylated-Rab8A, Rab10 and Rab12.

For the first time, our findings establish a connection between LRRK2 and cilia formation. Indeed, expression of activated LRRK2 mutant proteins interferes with ciliogenesis, a process that is influenced by both Rab8A and Rab10 proteins. Future studies are needed to shed light on the mechanistic details of how LRRK2 regulates the Rab-mediated transport of vesicles during ciliogenesis. In particular, it will be interesting to investigate whether the defective dopamine signaling in neurons of PD patients is at least partially due to LRRK2-mediated protein trafficking alterations in primary cilia. Although PD is not classified as a ciliopathy, subtle changes in ciliation may have profound signaling consequences in particular cell types that are critical for PD pathogenesis.

## Materials and methods

### Reagents

MLi-2 was synthesized by Natalia Shpiro (University of Dundee) as described previously (*Fell et al., 2015*). HG-10-102-01 was from Calbiochem. Doxycycline, γ-S-GTP, HA-agarose and trypsin from Sigma and LysC from Wako. GluC, AspN and Chymotrypsin from Promega. GFP-agarose beads were from Chromotek. Complete protease and phosphatase inhibitor tablets were from Roche.

## Antibodies

Anti-GFP (#2555), anti-GAPDH (#5174), anti-HA (#3724), anti-Rab1A (#13075), anti-Rab8A (#6975), and anti-Rab10 (#8127) were from Cell Signaling Technologies. Rab3A (PA1-770) and Rab5A (PA5-29022) from Thermo. Rab3B from Abnova (H00005865-M02). Rabbit monoclonal antibodies for total LRRK2 and pS935-LRRK2 were purified at the University of Dundee (*Dzamko et al., 2012*). Anti-Arl13B from Neuromab, goat anti-mouse Alexa 555 from Thermo Scientific. Rabbit monoclonal anti-Rab8A was from Abcam (ab188574). Sheep polyclonal antibodies against Rab29 phospho-Thr71 (S877D), Rab29 phospho-Ser72 (SA136) and Rab12 phospho-Ser106 (S876D) were generated by injection of KLH (keyhole limpet hemocyanin) and bovine serum albumin-conjugated phospho-peptides CIAGQERFT*SMTRLYYR (Rab29 pT71), CDIAGQERFTS*MTRLYYRSS (Rab29 pS72) or CAG-QERFNS*ITSAYYR (Rab12 pS106) into sheep and affinity-purified from serum using the same phosphopeptides. Antibodies were used for immunoblotting at final concentrations of 1 µg/ml, in the presence of 10 µg/ml of non-phosphorylated peptide. Sheep polyclonal antibodies against total Rab12 (SA227), Rab35 (SA314) and Rab43 (SA135) were generated by injection of recombinant full-length Rab12 (no epitope tag), GST-Rab35 or GST-Rab43 proteins into sheep and affinity-purified using the proteins with a different tag (6HIS-SUMO-Rab12, MBP-Rab35, HIS-SUMO-Rab43). Antibodies were used for immunoblotting at the final concentration of 1 µg/ml.

The polyclonal phospho-Rab8 antibody that efficiently immunoprecipitated multiple LRRK2 phosphorylated Rab proteins with a Thr site was from Abcam (Burlingame, California). The antibody was raised against two phospho-T72-Rab8A/Rab8B peptides (C-Ahx-AGQERFRT*ITTAYYR-amide and Ac-AGQERFRT*ITTAYYR-Ahx-C-amide, corresponding to residues 65–79 of human Rab8). The antibody was affinity purified employing the phospho peptide as a positive selection and the non-phosphorylated C-Ahx-AGQERFRTITTAYYR as a negative selection step. Rabbit monoclonal antibodies for phospho-T72 Rab8A/Rab8B and phospho-T73 Rab10 were custom-made by Abcam in collaboration with the Michael J Fox Foundation (Burlingame, California) (*Lis et al., 2017*). The phospho-Rab10 antibody was raised against two peptides (C-Ahx-AGQERFHT*ITTSYYR-amide and Ac-AGQERFHT*ITTSYYR-Ahx-C-amide, corresponding to residues 66–80 of human Rab10). The phospho-Rab8 antibody was raised against two peptides (C-Ahx-AGQERFRT*ITTAYYR-amide and Ac-AGQERFRT*ITTAYYR-Ahx-C-amide, corresponding to residues 65–79 of human Rab8). For immunization, the peptides were coupled to KLH via the Cys residue. *demark the phosphorylated residues.

## Plasmids

GFP-Rab7A (Addgene plasmid #12605) and GFP-Rab9A (Addgene plasmid #12663) were from Richard Pagano. GFP-Rab6A (Addgene plasmid #49469), GFP-Rab27A (Addgene plasmid #49605), GFP-Rab2A (Addgene plasmid #49541), GFP-Rab4A (Addgene plasmid #49434), GFP-Rab18 (Addgene plasmid #49550), GFP-Rab30 (Addgene plasmid #49607), GFP-Rab3A (Addgene plasmid #49542), GFP-Rab35 (Addgene plasmid #49552), GFP-Rab1A (Addgene plasmid #49467), GFP-Rab13 (Addgene plasmid #49548), GFP-Rab5A (Addgene plasmid #49888) and GFP-Rab10 (Addgene plasmid #49472) from Marci Scidmore. The following Rab constructs were from the University of Dundee: HA-Rab1B (DU25771), HA-Rab1C (DU55042), HA-Rab2B (DU51392), HA-Rab3A (DU51539), HA-Rab3B (DU55007), HA-Rab3C (DU55048), HA-Rab3D (DU26388), HA-Rab4B (DU51607), HA-Rab5A (DU26389), HA-Rab5B (DU26106), HA-Rab5C (DU47956), HA-Rab6B (DU47954), HA-Rab7B (DU54134), HA-Rab8A (DU35414), HA-Rab8B (DU39856), HA-Rab9B (DU54135), HA-Rab10 (DU44250), HA-Rab12 (DU48963), HA-RAB12-S106A (DU48966), GFP-Rab15 (DU26357), HA-Rab17 (DU54205), GFP-Rab26 (DU47921, DU47922), HA-Rab27B (DU53133), HA-Rab29 (DU50222), GFP-Rab31 (DU28238), HA-Rab32 (DU52622), HA-Rab33A (DU28313), HA-Rab33B (DU28314), HA-Rab35 (DU26478), GFP-Rab37 (DU53072), HA-Rab38 (DU52517), GFP-Rab39A (DU51554), GFP-Rab39B (DU54141), HA-Rab40A (DU53132), HA-Rab40B (DU28315), HA-Rab40C (DU53203), HA-Rab40AL (DU53222), GFP-Rab41 (DU28239), GFP-Rab43 (DU53044), HA-Rab43 (DU26392), GFP-Rab44 (DU53224), GFP-Rab45 (DU47924), HA-Rab8A-T72A/E (DU47360/DU47355), HA-Rab10-T73A/E (DU51006/DU51007), HA-Rab12-S106A/E (DU48966/DU48967), Flag-LRRK2-WT (DU6841), Flag-LRRK2-D2017A (DU52725), Flag-LRRK2-G2019S (DU10129), Flag-LRRK2-R1441G (DU13077), Flag-LRRK2-Y1699C (DU13165), Flag-LRRK2-Y1699C/D2017A (DU52703), RILP-GFP (DU27496), RILPL1-GFP (DU27305), RILPL2-GFP (DU27481), RILPL1[E280-END]-GFP (DU27317), RILPL1[E280-END,

R291L]-GFP (DU27478), RILPL1[E280-END, R291E]-GFP (DU27479), RILPL1[E280-END, R293L]-GFP (DU27471), RILPL1[E280-END, R293E]-GFP (DU27472), RILPL1[E280-END, R300L]-GFP (DU27473), RILPL1[E280-END, R300E]-GFP (DU27480), RILPL1[E280-END, K310L]-GFP (DU27474), RILPL1[E280-END, K310E]-GFP (DU27475, RILPL1[E280-END, K324L]-GFP (DU27476), RILPL1[E280-END, K324E]-GFP (DU27477), RILPL1[R291E] (DU27525), RILPL1[R293E] (DU27526), RILPL1[K310E] (DU27527), RILPL2[R130E] (DU27520), RILPL2[R132E] (DU27522), RILPL2[K149E] (DU27523) Full datasheets and reagents are available on https://mrcppureagents.dundee.ac.uk/. Mutagenesis on Rab3A/B/C/D, Rab5A/B/C, Rab8B, Rab29, Rab35 and Rab43 was carried out using Q5 site-directed mutagenesis kit (NEB, #E0554S).

## Culture and transfection of cells

HEK293, MEFs and NIH3T3 cells were cultured in Dulbecco's modified Eagle medium (Glutamax, Gibco) supplemented with 10% fetal calf serum (FCS), 100 U/ml penicillin and 100 µg/ml streptomycin. A549 cells were cultured in DMEM supplemented with 10% FBS, 2 mM L-glutamine, 100 units/ml penicillin and 100 µg/ml streptomycin. The HEK293-t-rex-flpIn stable cell lines with doxycycline-inducible mutant forms of LRRK2 have been described previously (Nichols et al., 2010). LRRK2 expression in HEK293-t-rex-flpIn was induced by doxycycline (1 µg/ml). NIH3T3 cells were transfected with eGFP-LRRK2-G2019S at 40% confluency and cilia formation induced by serum starvation after 24 hr. At the same time, 200 nM MLI-2 (or DMSO as control) was added and cells incubated for an additional 24 hr before fixation. LRRK2-R1441G MEFs were plated at 40% confluency and serum starved for 24 hr after 1 day. MLi-2 was added at 100 nM or 200 nM over night before fixation of the cells. Transient transfections were performed 24–48 hr prior to cell lysis using either FuGene 6, Fugene HD (Promega) or polyethylenimine PEI (Polysciences). hTERT-RPE cells were cultured in DMEM + F12 medium with 10% FBS. GFP-Rab8 (wt, T72A and T72E) were expressed using lentivirus vector pSLQ1371. After 48 hr of infection, cells were plated onto collagen coated coverslips and 48 hr later treated with serum-free medium to initiate ciliation. Cells were fixed with PFA and stained using mouse anti-Arl13B (Neuromab, 1:1000). All cells were tested for mycoplasma contamination and overexpressing lines were verified by western blot analysis.

## Generation of CRISPR-CAS9 knock-outs

The A549 Rab10 knockout cell line has been described previously (Ito et al., 2016). Rab8A, Rab35 and Rab43 knockout cell lines were generated using the following constructs (available at https://mrcppureagents.dundee.ac.uk): DU48928/DU48942 (Rab8A), DU54332/DU54340 (Rab35), DU54344/DU54350 (Rab43). Cells at about 80% confluency were co-transfected in a six-well plate with the pairs of corresponding vectors using Lipofectamine LTX, with a final amount of 9 µl Lipofectamine LTX and 2.5 µg DNA per well. 24 hr after transfection, the medium was replaced and fresh medium supplemented with puromycin (2 µg/ml). After 24 hr selection, the medium was replaced with medium without puromycin and the cells were left to recover for 48 hr before performing single-cell sorting using an Influx cell sorter (Becton Dickinson). Single cells were placed in individual wells of a 96-well plate containing DMEM supplemented with 10% FBS, 2 mM L-glutamine, 100 units/ml penicillin and 100 mg/ml streptomycin and 100 mg/ml normocin (InvivoGen). At about 80% confluency individual clones were transferred into six-well plates and screened for Rab8A, Rab35 or Rab43 knock-out by western blotting.

## Generation of mouse embryonic fibroblasts (MEFs)

LRRK2-R1441C knock-in mice were obtained from The Jackson Laboratory and maintained on a C57BL/6J background. Littermate matched wild-type and homozygous LRRK2-R1441C mouse embryonic fibroblasts (MEFs) were isolated from mouse embryos at day E12.5 resulting from crosses between heterozygous LRRK2-R1441C/WT mice as described (Wiggin et al., 2002). Cells were spontaneously immortalized by prolonged passaging. Genotyping of mice and MEFs was performed by PCR using genomic DNA. Primer 1 (5' -CTGCAGGCTACTAGATGGTCAAGGT −3') and Primer 2 (5' –CTAGATAGGACCGAGTGTCGCAGAG- 3') were used to detect the wild-type and knock-in alleles. The PCR program consisted of 5 min at 95℃, then 35 cycles of 30 s at 95℃, 30 s at 60℃ and 30 s at 72℃, and 5 min at 72℃.

## Immunofluorescence microscopy

Cells were fixed (3% paraformaldehyde, 15 min room temperature), permeabilized using Triton X-100 (0.1%), blocked with 2% BSA (in PBS) and stained with mouse anti-Arl13B and goat anti-mouse Alexa 555 antibodies (Both 1:1000). Nuclei were stained with DAPI (Sigma). Coverslips were mounted with Mowiol and NIH3T3 cilia visualized using an Olympus IX70 microscope with $60 \times 1.4$ NA Plan-Apochromat oil immersion objective (Olympus, Center Valley, PA) and a charge-coupled device camera (CoolSNAP HQ, Photometrics, Tucson, AZ). MEF-R1441G cells were visualized using an SP8 laser scanning confocal microscope (Leica Microsystems, Germany) using a $63 \times 1.4$ NA oil immersion objective. When needed, maximum intensity projections were made and images analyzed using ImageJ.

## Phos-tag SDS-PAGE and western blotting

Cells were lysed in ice-cold lysis buffer containing 50 mM Tris/HCl, pH 7.4, 1% (v/v) Triton X-100, 10% (v/v) Glycerol, 1 mM sodium orthovanadate, 50 mM NaF, 10 mM 2-glycerophosphate, 5 mM sodium pyrophosphate, 0.1 µg/ml mycrocystin-LR (Enzo Life Sciences, Switzerland) and complete EDTA-free protease inhibitor cocktail (Roche). Lysates were clarified by centrifugation at 20,800 g and then mixed with $4 \times$ SDS/PAGE sample buffer (250 mM Tris/HCl, pH 6.8, 8% (w/v) SDS, 40% (v/v) glycerol, 0.02% (w/v) Bromophenol Blue and 4% (v/v) 2-mercaptoethanol) and heated at 95°C for 5 min. For SDS/PAGE, 4–20 µg sample was loaded on to NuPAGE Bis-Tris 4–12% gel (Life Technologies) and electrophoresed at 150 V. For Phos-tag SDS-PAGE, samples were supplemented with 10 mM $MnCl_2$ before loading gels. Phos-tag SDS-Page was performed following the protocol described previously with slight modifications (*Ito et al., 2016*). Gels for Phos-tag SDS/PAGE consisted of a stacking gel [4% (w/v) acrylamide, 125 mM Tris/HCl, pH 6.8, 0.1% (w/v) SDS, 0.2% (v/v) *N,N,N',N'*-tetramethylethylenediamine (TEMED) and 0.08% (w/v) ammonium persulfate (APS) and a separating gel (12% (w/v) acrylamide, 375 mM Tris/HCl, pH 8.8, 0.1% (w/v) SDS, 100 µM Phos-tag acrylamide, 200 µM $MnCl_2$, 0.1% (v/v) TEMED and 0.05% (w/v) APS]. 4–12 µg of lysate or pull-down samples were electrophoresed at 70 V for the stacking gel and at 135 V for the separating gel with the running buffer [25 mM Tris/HCl, 192 mM glycine and 0.1% (w/v) SDS]. After SDS-PAGE, gels were washed three times with transfer buffer [48 mM Tris/HCl, 39 mM glycine and 20% (v/v) methanol] containing 10 mM EDTA and 0.05% (w/v) SDS, followed by a 10-min wash with transfer buffer containing 0.05% SDS. Blotting to nitrocellulose membranes was carried out at 90 V for 180 min on ice in the transfer buffer without SDS/EDTA. Transferred membranes were blocked with 5% (w/v) non-fat dry milk (NFDM) dissolved in TBS-T [20 mM Tris/HCl, pH 7.5, 150 mM NaCl and 0.1% (v/v) Tween 20] at room temperature for 45 min. Membranes were incubated with primary antibodies in 5% NFDM/TBS-T overnight at 4°C. After washing in TBS-T, membranes were incubated at room temperature for 1 hr with either horseradish peroxidase-labelled secondary antibody or with near-infrared fluorescent antibodies labeled with different fluorophores (680 and 800 nm, both diluted in 5% NFDM/TBS-T). After washing in TBS-T, protein bands were detected by exposing films [Medical Film (Konica Minolta)] to the membranes using an ECL solution [Amersham ECL Western Blotting Detection Reagents (GE Healthcare)] or by LI-COR scanning.

## Pull-downs

Cells were lysed either in ice cold NP-40 buffer (50 mM Tris-HCl, pH 7.5, 120 mM NaCl, 1 mM EDTA, 6 mM EGTA, 15 mM sodium pyrophosphate and 1% NP-40 supplemented with protease and phosphatase inhibitors [Roche]), in buffer containing γ–S-GTP (50 mM Tris-HCl, pH 7.5, 150 mM NaCl, 5 mM $MgCl_2$, 1 mM EDTA, 1 mM DTT, 10 mM sodium-glycerophosphate, 10 mM sodium pyrophosphate, 100 µM γ-S-GTP and 1% NP-40 supplemented with protease and phosphatase inhibitors) in case of endogenous Rab IPs or in 50 mM Tris/HCl, pH 7.4, 1% (v/v) Triton X-100, 10% (v/v) Glycerol, 1 mM sodium orthovanadate, 50 mM NaF, 10 mM 2-glycerophosphate, 5 mM sodium pyrophosphate, 0.1 µg/ml mycrocystin-LR (Enzo Life Sciences) and complete EDTA-free protease inhibitor cocktail.

Lysates were clarified by centrifugation at 14,000 rpm after a liquid nitrogen freeze-thaw cycle. For α-GFP and α-HA pulldowns, lysates were incubated with GFP/HA-agarose resin for 2 hr (20–30 µl beads/pulldown). Bound complexes were recovered by washing the beads twice with NP-40 buffer and twice with 50 mM Tris-HCl (pH 7.5) before overnight on-bead digestion at 37°C using

trypsin (~250 ng/pulldown in 2 M urea [dissolved in 50 mM Tris-HCl pH 7.5]). Digestion with enzymes other than trypsin (chymotrypsin, GluC, AspN) were performed according to manufacturer's instructions. The resulting peptides were processed as described in 'LC-MS-MS sample preparation'.

GFP-tagged RILP, RILPL1 and RILPL2 were affinity-purified from lysates (1–2 mg of protein) by incubation with 5 µl GFP-agarose resin (GFP-Trap_A, Chromotek) for 1 hr at 4°C. Immunoprecipitates were washed twice with lysis buffer and samples eluted from the beads by addition of 2 × SDS/PAGE sample buffer. The mixture was then incubated at 100°C for 10 min and the eluent collected by centrifugation through a 0.22 µm pore-size Spinex column before adding 2-Mercaptoethanol (1%). Samples were incubated for 5 min at 70°C prior to normal SDS-PAGE or Phos-tag analysis.

For immunoprecipitation of phospho-Rab from MEFs, the antibody was cross-linked to Protein-G-agarose beads at a ratio of 1 µg of antibody per 1 µl of beads using dimethyl pimelimidate as the cross-linking reagent. Lysates were incubated with the antibody-bead complexes for 2 hr at 4°C (20 µl of beads/2 mg of cell extract). The immunocomplexes were washed three times with PBS and samples eluted from the beads using 2 × SDS/PAGE sample buffer. The mixture was incubated at 100°C for 10 min and the eluent was collected by centrifugation through a 0.22 µm pore-size Spinex column before addition of 2-Mercaptoethanol (1%). Samples were incubated for 5 min at 70°C prior to western blotting.

## LC-MS/MS sample preparation

Peptides were purified using in-house prepared SDB-RPS (Empore) stage tips (*Rappsilber et al., 2003*) before LC-MS/MS analysis as described previously (*Kulak et al., 2014*). Briefly, stage tips were prepared by inserting two layers of SDB-RPS matrix into a 200 µl pipette tip using an in-house prepared syringe device. Stage-tips were first activated with 100 µl of 30% Methanol/1% Trifluoracetic acid (TFA) and then washed with 100 µl 2% Acetonitrile (ACN)/0.2% TFA before loading of the acidified peptides (1% TFA v/v). After centrifugation, the stage-tips were washed trice (200 µl each) with 2% ACN/0.2% TFA. Elution was performed using 60 µl of 60% ACN/1.25% $NH_4OH$. Eluates were collected in 200 µl PCR tubes and dried using a SpeedVac centrifuge (Eppendorf, Concentrator plus) at $60^0$C. Peptides were resuspended in buffer A* (2% ACN/0.1% TFA) and briefly sonicated (Branson Ultrasonics) before LC/MS-MS analysis.

## LC-MS/MS measurements

Peptides were loaded on a 20 or 50 cm reversed phase column (75 µm inner diameter, packed in-house with ReproSil-Pur C18-AQ 1.9 µm resin [Dr. Maisch GmbH]). Column temperature was maintained at 60°C using a homemade column oven. An EASY-nLC 1200 system (Thermo Fisher Scientific) was directly coupled online with the mass spectrometer (Q Exactive HF, Q Excative HF-X, LTQ Orbitrap, Thermo Fisher Scientific) via a nano-electrospray source, and peptides were separated with a binary buffer system of buffer A (0.1% formic acid [FA]) and buffer B (80% acetonitrile plus 0.1% FA), at a flow rate of 250 or 350 nl/min. Peptides were eluted with a nonlinear 45–180 min gradient of 5–60% buffer B (0.1% (v/v) formic acid, 80% (v/v) ACN). After each gradient, the column was washed with 95% buffer B for 5 min. The mass spectrometer was programmed to acquire in a data-dependent mode (Top10–Top15) using a fixed ion injection time strategy. Full scans were acquired in the Orbitrap mass analyzer with resolution 60,000 at 200 m/z (3E6 ions were accumulated with a maximum injection time of 25 ms). The top intense ions (N for TopN) with charge states $\geq$ 2 were sequentially isolated to a target value of 1E5 (maximum injection time of 120 ms, 20% underfill), fragmented by HCD (NCE 27%, Q Exactive) or CID (NCE 35%, LTQ Orbitrap) and detected in the Orbitrap (Q Exactive, R = 15,000 at m/z 200) or the Ion trap detector (LTQ Orbitrap).

*Figure 2* (Rab3B) and *Figure 2—figure supplement 1* (Rab5A): Xcalibur (v4.0.27.19) was used to build a method in which every full scan with resolution 120,000 at 200 m/z (3E6 ions accumulated with a maximum injection time of 20 ms) was followed by five data-dependent MS/MS scans with resolution 15,000 at 200 m/z (the target value of 1E5 with a maximum injection time of 32 ms, 1.4 m/z isolation window and NCE 27%) and a multiplexed SIM scan employed using the multiple ion injection method in which both light (endogenous) and heavy counterpart phosphopeptides of Rab5 and Rab3B were recorded simultaneously (amount of spike-in = 300 fmol per injection). Each SIM scan covered a range of m/z 150–2,000 with resolution 120,000 (1E5 ions accumulated with a

maximum injection time of 240 ms for both light and heavy counterparts, 1.4 m/z isolation window and 0.4 m/z isolation offset). For Rab5, m/z values of doubly (z = 2)- and triply (z = 3)-charged oxidized or non-oxidized YHpSLAPMYYR ions were defined as follows: 699.891, 691.894, 467.297, 461.965. For Rab3b, only the doubly (z = 2)-charged non-oxidized YRpTITTAYYR was targeted with an m/z value of 695.424.

LC-MS/MS analysis of pRab8 IPs (*Figure 2—figure supplement 2B*): Samples were analyzed using an Orbitrap Fusion Tribrid mass spectrometer coupled to Dionex Ultimate 3000 RSLCnano system (Thermo Scientific). Peptides were loaded on a pre-column (C18, 100A°, 2 cm PN: 164564) and resolved on 50 cm analytical column (C18, 100A°, 75µX 50 cm PN: ES803, Easy -Spray column, Thermo Scientific). Peptides were separated with a binary buffer system of buffer A (0.1% formic acid [FA]) and buffer B (0.1% (v/v) FA, 90% Acetonitrile [v/v]) by applying a non-linear gradient of 5% to 35% for 65 min and a total run of 85 min. Mass spectrometry data were acquired in a data-dependent mode targeting top 10 precursor peptides from the survey scan for MS2 fragmentation using HCD (32% NCE). Full scans were acquired in the Orbitrap mass analyzer with 120,000 resolution at 200 m/z and the MS/MS was acquired with Orbitrap with 30,000 resolution at 200 m/z. AGC was set at 2E5 and 5E4 for MS and MS/MS, respectively.

## Data processing and analysis

Raw mass spectrometry data were processed using MaxQuant version 1.5.3.15 or 1.5.8.3 (pRab8 IP of *Figure 2—figure supplement 2B*) (*Cox and Mann, 2008*; *Cox et al., 2011*) with an FDR < 0.01 at the level of proteins, peptides and modifications. Searches were performed against the Mouse or Human UniProt FASTA database (September 2015). Enzyme specificity was set to trypsin, GluC, AspN, LysC or chymotrypsin depending on the employed enzyme. The search included cysteine carbamidomethylation as a fixed modification and N-acetylation of protein, oxidation of methionine, and/or phosphorylation of Ser, Thr, Tyr residue (PhosphoSTY) as variable modifications. Up to two missed cleavages were allowed for protease digestion. Protein quantification was performed by MaxQuant (Label free quantification MaxLFQ [*Cox et al., 2014*]), 'match between runs' was enabled, with a matching time window of 0.7 min. Ion chromatograms of the SIM-MS files were extracted using Skyline (version 3.7.0.10940) (*MacLean et al., 2010*). Bioinformatic analyses were performed with Perseus (www.perseus-framework.org) and Microsoft Excel and data visualized using Graph Prism (GraphPad Software Inc) or Perseus (*Tyanova et al., 2016*). Significance was assessed using one sample t-test, two-sample student's t-test and ANOVA analysis, for which replicates were grouped, and statistical tests performed with permutation-based FDR correction for multiple hypothesis testing. Were indicated, missing data points were replaced by data imputation after filtering for valid values (all valid values in at least one experimental group). Error bars are mean ±SEM or mean ±SD.

## Acknowledgements

This work was supported The Michael J. Fox Foundation for Parkinson's Research (grant ID 6986.05), the Max-Planck Society for the Advancement of Science and the Medical Research Council (grant numbers MC_UU_12016/2 (to DRA). We thank S Uebel, M Spitaler and S Pettera from the MPIB Biochemistry Core Facility and S Kroiss, G Sowa, K Mayr and I Paron from the department of Proteomics and Signal Transduction for technical assistance. Members of the Dundee MRC-PPU Cloning, DNA Sequencing, antibody, protein production, tissue culture and mass spectrometry teams for technical assistance. Sandy Chou at Abcam for generation of the phospho-Rab8 polyclonal antibody. LRRK2 R1441G MEFs were kindly provided by Dr Shu-Leong Ho (Division of Neurology, Department of Medicine, University of Hong Kong) and have been described previously (*Ito et al., 2016*).

## Additional information

### Competing interests

Suzanne R Pfeffer: Reviewing editor, *eLife*. The other authors declare that no competing interests exist.

## Funding

| Funder | Grant reference number | Author |
|---|---|---|
| Max-Planck-Gesellschaft | | Martin Steger<br>Ozge Karayel<br>Matthias Mann |
| Michael J. Fox Foundation for Parkinson's Research | 6986.05 | Martin Steger<br>Federico Diez<br>Herschel S Dhekne<br>Pawel Lis<br>Raja S Nirujogi<br>Ozge Karayel<br>Francesca Tonelli<br>Terina N Martinez<br>Suzanne R Pfeffer<br>Dario R Alessi<br>Matthias Mann |
| Medical Research Council | 357811350 R60 | Federico Diez<br>Pawel Lis<br>Raja S Nirujogi<br>Francesca Tonelli<br>Dario R Alessi |

The funders had no role in study design, data collection and interpretation, or the decision to submit the work for publication.

## Author contributions

Martin Steger, Conceptualization, Validation, Investigation, Visualization, Methodology, Writing—original draft, Writing—review and editing; Federico Diez, Herschel S Dhekne, Investigation, Visualization, Methodology; Pawel Lis, Raja S Nirujogi, Ozge Karayel, Investigation, Visualization; Francesca Tonelli, Validation, Investigation; Terina N Martinez, Resources; Esben Lorentzen, Conceptualization, Writing—original draft; Suzanne R Pfeffer, Conceptualization, Supervision, Funding acquisition, Investigation, Visualization, Writing—review and editing; Dario R Alessi, Conceptualization, Supervision, Funding acquisition, Writing—original draft, Project administration, Writing—review and editing; Matthias Mann, Conceptualization, Supervision, Funding acquisition, Investigation, Methodology, Writing—original draft, Project administration, Writing—review and editing

## Author ORCIDs

Martin Steger http://orcid.org/0000-0003-1637-8190
Esben Lorentzen http://orcid.org/0000-0001-6493-7220
Suzanne R Pfeffer http://orcid.org/0000-0002-6462-984X
Dario R Alessi http://orcid.org/0000-0002-2140-9185
Matthias Mann http://orcid.org/0000-0003-1292-4799

## Decision letter and Author response

Decision letter https://doi.org/10.7554/eLife.31012.017
Author response https://doi.org/10.7554/eLife.31012.018

# Additional files

## Supplementary files

• Supplementary file 1 Summary of used proteases and identified peptides for the proteomic analysis of 52 Rab GTPases. Proteomics raw data have been deposited to the ProteomeXchange Consortium (*Vizcaíno et al., 2014*) via the PRIDE partner repository with the data set identifier PXD007214.
DOI: https://doi.org/10.7554/eLife.31012.014

• Transparent reporting form
DOI: https://doi.org/10.7554/eLife.31012.015

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
