## [Decision Letter]

Thank you for submitting your article "Systematic proteomic analysis of LRRK2-mediated Rab GTPase phosphorylation establishes a connection to ciliogenesis" for consideration by *eLife*. Your article has been reviewed by two peer reviewers, and the evaluation has been overseen by a Reviewing Editor and Ivan Dikic as the Senior Editor. The reviewers have opted to remain anonymous.

The reviewers have discussed the reviews with one another and the Reviewing Editor has drafted this decision to help you prepare a revised submission.

Summary:

In this manuscript, Steger et al. examined potential LRRK2 targets among the family of Rab GTPases, which the authors identified in their previous work on LRRK2 (Steger et al., *eLife* 2016). By employing phospho-proteomics in combination with Rab/LRRK2 overexpression and pharmacological LRRK2 inhibition, the authors validated 14 out of 50 Rabs as LRRK2 candidate substrates. Consistently, the phosphorylation status of all of these candidates were found to be affected by overexpression of LRRK2 variants carrying Parkinson disease (PD)-associated mutations, which increase the kinase activity of LRRK2. Steger and colleagues made an effort to validate potential kinase-substrate relationship at the endogenous level. Using Rab- and phospho-site-specific antibodies in combination with LRRK2-activating mutations and a specific LRRK2 inhibitor, the authors firmly established six out of 14 Rab proteins as endogenous LRRK2 phosphorylation targets. Next, the authors sought to address whether the LRRK2-mediated phosphorylation impacted on the protein-protein interaction landscape of the Rab substrates. By focusing on the previously established LRRK2 substrate Rab8A (Steger et al., e*Life* 2017), the authors identified RILPL2 as a novel phosphorylation-dependent association partner. Further binding and mutation experiments confirmed that LRRK2-phosphorylated Rab8A binds to the RH domain of RILPL2. Consistent with the involvement of Rab8A and RILPL2 in ciliogenesis, the authors found that pathogenic LRRK2 mutations affect primary cilia formation.

This is an important study that provides a real new insight into processes that may contribute to the aetiology of PD. Like most really good studies it raises almost more questions than it answers, especially the exact molecular mechanisms by which Rabs and RILPLs function in ciliogenesis, the consquences of defects in their activity, and the role of the network in disease.

Together, work in this manuscript elegantly combines well-controlled proteomics and biochemical experiments and provides an excellent and expedient extension of the author's previous work on LRRK2.

1) Is RLPL2 found in the proteomics analysis of Rab8a (T72A) and Rab8a (T72E) shown in Figure 5 of Steger et al., *eLife* 2016. If that is the case, the authors might want to indicate this.

2) The author's use of the term "isoform" throughout the manuscript is not clear. For example in the Results and Discussion the authors state "[…]in 37 different Rab proteins. For 12 isoforms[…]". I would not consider Rab3A and Rab5b to be isoforms (in contrast to e.g. Rab3a and Rab3b). The authors should rephrase the respective sentences.

3) Subsection “LRRK2 phosphorylates at least six endogenous Rab GTPases in cells” (Figure 2): it is not clear how endogenous Rab8A, Rab10 and Rab12 are immunoprecipitated. I assume by specific antibodies. But this is not mentioned in the main text or figure legend. Please add this information.

4) The described SIL method sounds a lot like the absolute quantification (AQUA). The authors should acknowledge this established technique. If SIL is considerable different that AQUA that it would be worth describing this.

5) The authors describe the use of an antibody against LRRK2-phosphorylated Rab8A but subsequently this antibody was named either phospho-Rab8 antibody or pan-selective Rab phospho-antibody. An unifying nomenclature would make the reader's life easier.

6) Figure 4: Is the effect of hyper-activated LRRK2 on cilia formation mediate by LRRK2 phosphorylation of Rab8A and/or modulated binding of Rab8A and RILPL2? The authors should explore this by employing Rab8A phospho-mimetic and phospho-ablating mutations (in particular since these tools are already available in the authors' labs).

---

## [Author Response]

[…]1) Is RLPL2 found in the proteomics analysis of Rab8a (T72A) and Rab8a (T72E) shown in Figure 5 of Steger et al., eLife 2016. If that is the case, the authors might want to indicate this.

No, this was not the case and it was very surprising for us, too. We think that the substitution of Rab8a-T72 to a glutamic acid residue does not effectively mimic for the phosphorylation induced by LRRK2 in this case. We discuss this in our conclusions.

We also included an additional confirmation of the phospho-dependent Rab8a-RILPL1/2 interaction at endogenous expression levels in LRRK2-R1441C MEFs (Figure 3).

2) The author's use of the term "isoform" throughout the manuscript is not clear. For example in the Results and Discussion the authors state "[…]in 37 different Rab proteins. For 12 isoforms[…]". I would not consider Rab3A and Rab5b to be isoforms (in contrast to e.g. Rab3a and Rab3b). The authors should rephrase the respective sentences.

We agree that the usage of the ‘isoform’ term is confusing and we changed this to ‘Rabs’ or ‘Rab proteins’ throughout the manuscript.

3) Subsection “LRRK2 phosphorylates at least six endogenous Rab GTPases in cells” (Figure 2): it is not clear how endogenous Rab8A, Rab10 and Rab12 are immunoprecipitated. I assume by specific antibodies. But this is not mentioned in the main text or figure legend. Please add this information.

We apologize for not including this information! We slightly modified the text as follows:

“For this, we overexpressed the pathogenic LRRK2-Y1699C form in HEK293 cells, immunoprecipitated endogenous Rab proteins using commercially available- (Rab3A, Rab3B, Rab5A, Rab8A, Rab10) or in-house raised antibodies (Rab12) and quantified LRRK2-mediated phosphorylation by LC-MS/MS.”

All detailed information about the antibodies can be found in Materials and methods section.

In this context, we also investigated the phosphorylation of other Rabs by LRRK2 in more detail. Probably because of its low expression levels in HEK293 cells, we were not able to detect any LRRK2 mediated phosphorylation of Rab3B by employing a standard (data-dependent) mass spectrometry method. However, using our sensitive method, we found that endogenous Rab3B is clearly phosphorylated by overexpressed LRRK2 in HEK293 cells (now included in Figure 2).

Also, similar to the IP presented in Figure 2—figure supplement 2, we used the phospho-Rab8 antibody to selectively enrich phosphorylated Rabs from LRRK2-Y1699C expressing HEK293 cells. Strikingly, in this setup we were able to quantify a number of endogenous Rabs that we had also identified in our initial overexpression screen, and these were Rab3A/B/C/D, Rab8A/B, Rab10, Rab35 and Rab43. This result shows that LRRK2 can phosphorylate these Rabs when expressed at endogenous levels and we therefore changed Abstract, the section ‘LRRK2 phosphorylates at least ten endogenous Rab GTPases in cells’ and corresponding parts of Figure 2 accordingly.

4) The described SIL method sounds a lot like the absolute quantification (AQUA). The authors should acknowledge this established technique. If SIL is considerable different that AQUA that it would be worth describing this.

The method we use for targeted detection of low abundant Rab phosphopeptides has indeed some similarities with the ‘absolute quantification (AQUA)’ strategy published previously and we included one sentence to acknowledge this:

“Therefore, to further increase mass spectrometric sensitivity, we used stable isotope labeled (SIL) peptides equivalent to the endogenous Rab peptides of interest to guide targeted quantification. In contrast to the absolute quantification (AQUA) strategy (Gerber, Rush et al., 2003) that relies on fragment ions for quantification, we recorded both light and heavy counterparts in a multiplexed selected ion monitoring (SIM) scan mode on a quadrupole Orbitrap analyzer and used the heavy form to indicate the retention time (heavy) and the light form for quantification.”5) The authors describe the use of an antibody against LRRK2-phosphorylated Rab8A but subsequently this antibody was named either phospho-Rab8 antibody or pan-selective Rab phospho-antibody. An unifying nomenclature would make the reader's life easier.

The described antibody was raised against the linear sequence surrounding pT72 of both Rab8a and Rab8b (the primary amino acid sequence is identical in both proteins). We therefore named it phospho-Rab8 antibody. Since our experiments revealed that the antibody actually recognizes a number of LRRK2 phosphorylated Rabs with a Thr site in their switch-II domain, we referred to it as pan-selective phospho-Rab antibody. However, we acknowledge that the use of this nomenclature was confusing and we name it ‘phospho-Rab8 (or pRab8) antibody ‘throughout the manuscript.

6) Figure 4: Is the effect of hyper-activated LRRK2 on cilia formation mediate by LRRK2 phosphorylation of Rab8A and/or modulated binding of Rab8A and RILPL2? The authors should explore this by employing Rab8A phospho-mimetic and phospho-ablating mutations (in particular since these tools are already available in the authors' labs).

Thanks to the reviewer’s suggestions, we investigated RILPL1/2 Rab interaction in depth and the results of this were remarkable. In detail, we first performed an unbiased AE-MS experiment using GFP-RILPL1 and GFP-RILPL2 (both wt and the described R→E Rab interaction mutants) and found that not only Rab8A, but also Rab10 interacted with RILPL1 in a LRRK2 specific manner. RILPL2 instead interacted with pRab8A, pRab10 and pRab12 in this setup. The results of this are shown in Figure 3. We independently confirmed the phospho-specific interaction by immunoprecipitation/phos-tag SDS-PAGE (Figure 3—figure supplement 3) and moreover confirmed that several R→E or K→E mutants in both full-length RILPL1 and RILPL2 mediate pRab interaction (Figure 3).

Second, as proposed by the referees, we analyzed the effect of Rab8a-T72A and Rab8A-T72E overexpression on cilia formation. Surprisingly, we did not find any effect on ciliogenesis when overexpressing these mutant proteins in RPE-1 cells and contrary to the wt, both T72A and T72E mutants did not localize at cilia. This demonstrates that overexpression of the mutants alone is not sufficient to interfere with ciliogenesis and further investigation is needed to determine which other factors might be required for this. The data is shown in Figure 4.